# Colloidal Stability of Silica-Modified Magnetite Nanoparticles: Comparison of Various Dispersion Techniques

**DOI:** 10.3390/nano11123295

**Published:** 2021-12-04

**Authors:** Gulzhian Dzhardimalieva, Lyubov Bondarenko, Erzsébet Illés, Etelka Tombácz, Nataliya Tropskaya, Igor Magomedov, Alexander Orekhov, Kamila Kydralieva

**Affiliations:** 1Department of General Engineering, Moscow Aviation Institute, National Research University, 125299 Moscow, Russia; dzhardim@icp.ac.ru (G.D.); l.s.bondarenko92@gmail.com (L.B.); ntropskaya@mail.ru (N.T.); eaua2@yandex.ru (I.M.); lsk.orekhov@gmail.com (A.O.); 2Laboratory of Metal Polymers, Institute of Problems of Chemical Physics, 142432 Chernogolovka, Moscow Region, Russia; 3Department of Food Engineering, University of Szeged, 6720 Szeged, Hungary; Illes.Erzsebet@chem.u-szeged.hu; 4Soós Ernő Water Technology Research and Development Center, University of Pannonia, 8800 Nagykanizsa, Hungary; e.tombacz@chem.u-szeged.hu; 5Sklifosovsky Institute for Emergency Medicine, 129090 Moscow, Russia

**Keywords:** silica-modified magnetite nanoparticles, dispersion, zeta-potential, hydrodynamic size, polyanions loading

## Abstract

The production of stable and homogeneous batches during nanoparticle fabrication is challenging. Surface charging, as a stability determinant, was estimated for 3-aminopropyltriethoxysilane (APTES) coated pre-formed magnetite nanoparticles (MNPs). An important consideration for preparing stable and homogenous MNPs colloidal systems is the dispersion stage of pre-formed samples, which makes it feasible to increase the MNP reactive binding sites, to enhance functionality. The results gave evidence that the samples that had undergone stirring had a higher loading capacity towards polyanions, in terms of filler content, compared to the sonicated ones. These later results were likely due to the harsh effects of sonication (extremely high temperature and pressure in the cavities formed at the interfaces), which induced the destruction of the MNPs.

## 1. Introduction

Magnetic nanoparticles (MNPs) modified with silane have a wide range of applications, as nanocarriers for drug delivery systems in biomedicine [1], as sorbents for removal ecotoxicants in ecology [2], as agents for magnetorheological fluids [3,4,5], and in particular, as fluid shock absorbers for aircrafts [6,7], etc. The production of homogeneous batches during nanoparticle (NP) fabrication is challenging. However, the fabrication of Fe_3_O_4_ MNPs modified by a SiO_2_ via self-assembly silylation method results in higher heterogeneity, because of the complex thermodynamic processes (the silylation process involves more than a single mechanism [8]) and kinetics involved. The size of the NPs can increase due to aggregation. This can lead to a decrease in NP mobility due to deposition mechanisms such as attachment, deformation, and maturation [9,10]. The sedimentation and aggregation of NPs can lead to a decrease in the efficiency of their application for magnetorheogical fluids. Consequently, the stability of nanoparticle suspensions plays a pivotal role in magnetorheogical fluid development.

The method of dispersion selected affects the level of agglomeration, altering the charge state and hydrodynamic diameter of NPs in solution [11]. The tools selected for dispersion of NPs in suspensions will lead, either to different degrees of dissolution, agglomeration, and transformation into other phases, or to various typologies of interactions with environmental components [12]. In this regard, understanding the transformation of nanoparticles as a result of processing is a very important topic in any field of research where nanoparticle dispersions are used. Therefore, knowledge of the impact of sample formulation on the particulate specifics (e.g., size, surface active sites, and zeta potential) and dissolved fraction is very important [13,14], as this procedure will have a substantial influence on the bioactivity of NPs [15].

There are several protocols for the preparation of NP dispersions [16,17,18,19,20]. Typically, these protocols involve either applying specific acoustic energy to the solution or a sufficiently long sonication time, which will not reduce the size of the aggregates with increasing sonication time [12]. Accessible standard dispersion protocols have been developed for nanoparticles with a very slow rate of dissolution/transformation, such as TiO_2_ and SiO_2_ [13,14]. However, such levels of energy may be sub-optimal for other types of nanoparticles, and, therefore, the surface and dissolution properties may change [21], thereby, affecting the interaction of MNPs with contaminants [22]. The follow-up dispersal settings will also continue to be of utmost importance. The long-term dilution of nanoparticles by sonication treatment, for example, can lead to the presence of both nanoparticles and released ions in the final suspension [23]. This should be taken into account; e.g., as the surface charge, the chemical speciation of released metal species, and their combinations influence the sorption capacity of nanoparticles [24].

3-Aminopropyl-triethoxysilane (APTES) is the most commonly used alkoxysilane, due to its terminal amino groups [8,25,26,27]. APTES, as an SiO_2_-derivative, is stable under acidic conditions and inert to redox reactions, compared with organic coating materials, and hence functions as an ideal shell composite to protect the inner magnetite core from aggregation and sedimentation [28]. Extra-surface modification of Fe_3_O_4_, utilizing APTES via silylation reaction, allows keeping the Fe_3_O_4_ chemical stability, thanks to strong covalent Fe-O-Si bond formation, on the one hand, while, on the another hand, the non-hydrolyzable amino groups of alkoxysilanes can be reactive [29]. 

This work shows how widely used routine preparation methods affect the surface charge and hydrodynamic diameter of synthesized nanoparticles. Thus, the aim of the present manuscript was mainly focused on studying the potentiality of dispersion effects that can alter (*i*) the stability of dispersed nanomaterials and (*ii*) the loading capacity of model polyanions-humic substances playing the role of a functional component; for example, in processes of water cleaning from heavy metals [30,31] and organic contaminants [32,33], as well as antibacterial, antiallergenic, and anti-inflammatory agents [34,35]. Changes in zeta-potential after standard dispersion techniques (i.e., stirring and sonication) in silica-coated magnetite aggregates were also reported and analyzed. 

## 2. Materials and Methods

### 2.1. Synthesis of Fe_3_O_4_ MNPs

Unmodified Fe_3_O_4_ nanoparticles were obtained by coprecipitation, according to the Elmore reaction [36]. Subsequently, 7.56 g of iron (III) chloride and 2.78 g of iron (II) chloride were suspended in 70 mL of H_2_O, then 40 mL of 25% ammonium hydroxide solution was added at 50 °C, with intensive stirring at 1000 rpm in argon. To remove unreacted synthesis residues, bare Fe_3_O_4_ nanoparticles were washed five times with ultrapure water and then dried at 70 ° C in vacuum.

### 2.2. Synthesis of Fe_3_O_4_/APTES MNPs

Previously prepared bare magnetite nanoparticles were coated with aminoorganosilane-3-aminopropyltriethoxysilane (APTES). The synthesis of the silica-coated Fe_3_O_4_ nanoparticles was carried out by sol-gel synthesis, and 3-aminopropyltriethoxysilane (APTES, 98% purity, Sigma-Aldrich, Merck KGaA, Darmstadt, Germany) was used as the silica precursor, performing the synthesis in argon atmosphere. In accordance with [37], 3.2 g of unmodified Fe_3_O_4_ was supported by 150 mL of ethanol solution/water with a volume ratio of 1:1, then 13.6 g of APTES was added to the solution for 2 h in an argon atmosphere at 40 °C. A molar ratio of APTES to Fe_3_O_4_ of 4:1 was employed. The room temperature-cooled solution of Fe_3_O_4_/APTES MNPs was separated with a magnet (Nd, 0.3 T), washed sequentially with ethanol, and distilled water three times, removing the by-products. Finally, the Fe_3_O_4_/APTES MNPs were dried in a vacuum at 70 °C for 2 h. The Fe_3_O_4_/APTES sample was purified using dialysis for 2 weeks against ultrapure water and referred to as Fe_3_O_4_/APTES.

### 2.3. Dispersion by Sonication 

The Fe_3_O_4_/APTES was dispersed using sonication in an ultrasonic bath (30 kHz, 50 W) for 30 min and referred to as Fe_3_O_4_/APTES-Us.

### 2.4. Dispersion by Stirring 

The Fe_3_O_4_/APTES sample was processed for 24 h on a magnetic stirrer at 600 rpm and referred to as Fe_3_O_4_/APTES (S).

In total, four types of sample were analyzed, respectively, Fe_3_O_4_, Fe_3_O_4_/APTES, Fe_3_O_4_/APTES (Us), and Fe_3_O_4_/APTES (S) in terms of surface charging, hydrodynamic diameter, and loading capacity towards humic preparation (HP).

### 2.5. Humic Preparation Characterization

The humic preparation was obtained from sodium humate (Powhumus, Humintech, Grevenbroich, Germany). The low H/C atomic ratio of 0.85 indicates a high content of aromatic structures in the humic preparation. The HP contains 5.3 mmol/g of acidic COOH and OH-groups, with an average molecular weight, Mw, of 9.9 kD.

#### Characterization of Samples

The phase composition and particle size of unmodified and modified magnetic nanoparticles were determined by means of X-ray diffraction (XRD) (Philips X’Pert diffractometer; Philips, Amsterdam, the Netherlands, Cr-Kα, λ = 2.29 106 Å). The particle size was determined using the Sherrer equation taking into account the full width at half maximum (FWHM) of all peaks. To estimate the content of stoichiometric magnetite, after fabrication and modification, the reflection (i.e., 440) was fitted with five different mathematical functions (a Gaussian, a Lorentzian, a Voigt, a Pseudo-Voigt, and a Pearson VII function) in Origin 2019 Pro.

Acid-base titration was performed in a custom automatic device consisting of a titration vessel, a magnetic stirrer, a nitrogen gas inlet, burettes (Metrohm, Dosimat 665, Metrohm AG, Herisau, Switzerland) for acid and base standard solutions, a glass electrode (OP-0808P by Radelkis Ltd., Budapest, Hungary) for pH measurement, and a high precision potentiometer (made at the University of Szeged, Szeged, Hungary). The electrodes were treated according to the storage, cleaning, and activation instructions of the manufacturer. The measurement accuracy of the potentiometer was ±0.01 mV. The titration was controlled by custom software (Gimet-1, homemade software, Szeged, Hungary), applying the equilibrium method [38,39,40,41].

Dynamic light scattering (DLS) measurements were performed using a Nano ZS apparatus at 25 °C and 633 nm wavelength with a solid-state He-Ne laser having a scattering angle of 173°. All samples obtained were diluted to about 0.1 g/L. Each sample was measured at a given kinetic state obtained after 10 s of sonication, to avoid aggregation and achieve the same colloidal state for the suspensions. The measurements were performed in a zeta cell (DTS1070, Malvern Panalytical Ltd., Malvern, UK) over a predetermined pH range of ~3 to ~10. The pH values were adjusted before measurement and checked after the study. The experiments were carried out at a constant ionic strength of 0.01 M NaCl. The effect of polyanions loading on the surface capacity of MNPs was tested.

## 3. Results

### 3.1. Characterization of MNP Microstructure

According to the X-ray diffractometry studies, the major phase formed in the present and absence of the APTES was magnetite Fe_3_O_4_. The lattice constant calculated from XRD decreases after modification from 8.38 Å (Fe_3_O_4_) to 8.37 Å (Fe_3_O_4_/APTES) (Table 1).

The composition of the crystalline component of the samples can be allocated as follows using the approach in [39]. Fe_2.94_O_4_ and Fe_2.88_O_4_ for the Fe_3_O_4_ and Fe_3_O_4_/APTES samples, respectively (Table 1). The changes in modified magnetite stoichiometry were likely caused by the oxidation of magnetite during APTES deposition.

The SEM images [40] show greater particle diameters compared with the XRD, whereas SEM allows measuring the size of aggregates formed from either crystalline or amorphous nanoparticles. The diameters measured for Fe_3_O_4_ nanoparticles were typical for superparamagnetic MNPs with high saturation magnetization and a high specific surface area [41]. According to the SEM data (Table 1), the particle diameter upon APTES modification did not change as compared to bare magnetic nanoparticles. All samples were considered to be polydisperse [42]. FTIR-spectroscopy of Fe_3_O_4_ and Fe_3_O_4_-APTES from our previous paper [40] confirmed the following: (a) formation of covalent bonds between the magnetite core and the silanol shell Fe-O-Si; (b) polymerization with the formation of bonds Si-O-Si; and (c) presence of NH_2_ amino groups on the surface, which were ideal for further modification.

### 3.2. Characterization of the Zeta Potential and Hydrodynamic Size of MNPs

#### 3.2.1. Surface Charging of MNPs

Potentiometric acid–base titration was performed in the pH range 3–11 at 0.005, 0.05, and 0.5 M KCl concentrations [43]. Magnetite particles can develop charges due to the protonation and deprotonation reactions of ≡Fe-OH surface sites, such as ≡Fe-OH + H^+^ < = > ≡Fe-OH_2_^+^ and ≡Fe-OH + OH^-^ < = > ≡Fe-O^-^ + H_2_O, respectively. 

H+-ions accumulate on the surface at pHs lower than the point of zero charge (PZC); therefore, the surface charge is positive, while magnetite particles are negatively charged in alkaline solutions above the pH of the PZC. The experimental PZC value of magnetite was pH~7.8. The surface charge density increased with increasing KCl concentration due to the charge screening effect of the salt.

Below pH~3 and above pH~10.5, the dissolution of metal oxides may occur. The solubility of magnetite, owing to its Fe (II) content, is usually higher than that of pure Fe (III) oxides. Nevertheless, the concentration of dissolved Fe (III) species is small (i.e., not greater than ~10^−5^ mol/dm^3^ within pH range from ~4 to ~10 [44]), and, thus, the dissolution of particles can be ignored within the titration pH range.

When comparing the curves of direct and reverse potentiometric titration of the Fe_3_O_4_/APTES sample (0.005 M and 0.05 M KCl), a hysteresis loop was observed among the curves (refers to Figure 1), likely due to the presence of impurities or by-products.

Furthermore, in this study we used dialysis for purification, i.e., removal of excess ligand, salt, starting material, by-products. Dialysis consisted in immersing a semipermeable bag with a nanomaterial dispersion in a receptor solution [45]. This method is beneficial in terms of the minimal manipulation of the sample, without the need for any pretreatment. Routine tools for NPs dispersion such as bath sonication and magnetic stirring [19,45] were compared.

#### 3.2.2. Effects of Dialysis and Dispersion

Dialysis is a relatively cheap, commercially available, simple to use, and gentle purification method for removing the excess of reagents after the deposition of chemical groups on a particle surface, although it is time-consuming [46]. The driving force behind dialysis is the different concentration of the solute inside and outside of a membrane. Electro-kinetic measurements allow estimating the electric double layer of charged particles. The measured electrophoretic mobility value can be transformed into the electrokinetic (zeta) potential value, taking into account the known theoretical basis with several assumptions, restrictions, and remarks. The pH of the isoelectric point (IEP) can be identified as a characteristic pH, at which the sign of the electrophoretic mobility of MNPs dispersion reverses. The symmetric shape of the zeta potential–pH line near the IEP indicates the importance of the amount of H^+^/OH^–^ ions in determining surface charging level [47]. When assessing the surface charging of modified magnetite samples, it was revealed that treatment of the samples affects the zeta potential value of MNPs.

The zeta potential curves are reported in Figure 2 for the Fe_3_O_4_/APTES samples after extra treatment, including purification by using dialysis and dispersion based on sonication and stirring techniques. The presence of amino groups in silica-MNPs [48,49] shifts the IEP towards a higher pH value (7.1) (Figure 2a) in comparison to Fe_3_O_4_ with the IEP ca. 6.3 [50]. This is combined with the position of the ζ-potential curve, whereby there are less negative charges in the region after pH~7 (alkaline region) on the Fe_3_O_4_/APTES surface than on Fe_3_O_4_. The protonation of Fe_3_O_4_-Si–NH_2_ surface sites with H+ leads to the formation of positive (FeO-Si-${\mathrm{NH}}_{3}^{+}$) surface charges.

According to [49], a zeta potential value of +30 mV in the entire pH range indicates the stability level of an aminosilica-coated nanocomposite. In addition, this may also indicate the presence of NH_3_^+^ groups on the surface of MNPs [50]. The Fe_3_O_4_/APTES MNPs are stable in the pH range 3–5, beyond which the ζ-potential starts to decrease and the system becomes less stable. As a result, at pH > 7, the MNPs becomes unstable, with a ζ-potential of about –10 mV. This behavior is in good agreement with the value of the hydrodynamic diameter in this range (about 250 nm) (Figure 2b).

Upon dialysis of Fe_3_O_4_/APTES, the IEP value, which indicates a change in the surface charging of samples shifted in the case of samples which had experienced different dispersion treatments (Figure 2a, Table 2). The final results revealed that the IEP varied from 7 to 6.6, respectively, for the untreated and dialysis-stirrer-treated samples, with a further reduction to 6.3 in the case of the dialyzed-sonicated systems

#### 3.2.3. Loading Capacity towards Humic Preparation

The effect of humic preparation (HP) adsorption on the ζ-potential of MNPs was studied and the results are reported hereafter. Due to the polyfunctionality of humic substances, they are capable of almost any kind of chemical interaction: ionic, donor-acceptor, hydrophobic bonding. As a result, humic substances form complexes with metals and bind both highly hydrophobic and functionalized organic compounds. Humic substances-coated Fe_3_O_4_ nanoparticles have wide applicability in the removal of ecotoxicants, such as toxic metals Hg (II), Pb (II), Cd (II), and Cu (II); arsenic and chromium, radionuclides, and aromatic compounds from various waters [51,52].

The content of HP was increased, to achieve entirely covered MNPs. The influence of different HP content is presented in Appendix A, considering different ratios of HP vs. Fe_3_O_4_/APTES, namely 0.0077–0.1161 g/g (Appendix A).

The Figure 3 shows the difference between the samples before dialysis (Fe_3_O_4_/APTES), which were subjected to sonication and the samples after dialysis subjected to different pre-treatments (Fe_3_O_4_/APTES (Us) and Fe_3_O_4_/APTES (S)) taking into account uncertainty ± at the sorption of 0.026 g/g humic preparations. Therefore, in the interval pH 3–5, the use of dialysis leads to an increase in the positive charge of Fe_3_O_4_/APTES, probably, due to the removal of impurities that screened for NH_3_^+^ on the surface. The use of a magnetic stirrer after dialysis in the case of Fe_3_O_4_/APTES (S) led to a significant increase in the positive surface charge (from −16.3 mV to 17.2 mV at pH 3) compared to the sample not subjected to dialysis (Fe_3_O_4_/APTES). At the same time, the use of a magnetic stirrer in comparison with sonication under these conditions also led to an increase in the positive surface charge and, consequently, in the sorption capacity in relation to the selected polyanion in the range of pH 3–6.

Under specific concentrations, the HP can entirely cover the MNP surface (Appendix A). According to [43], humic preparations have a negative charge in the entire pH range studied here. The adsorption of a negatively charged polyanionic humic preparation on MNPs leads to a recharging of the acidic region surface, from positive to negative; negative charges on nanoparticles gradually become predominant. The attainment of a negative surface charge in the entire pH range (i.e., 3–10) indicates the complete coverage of the HP surface by the humic preparation (Figure 4). The maximum value of negative zeta potential achievable with an increase in HP concentration is about −40 mV.

In the case of partial coating by HP preparation, nanoparticles had patch-wise charging heterogeneity and they stuck together, inducing a unstable dispersion, as indicated by the zeta potential falling within the range + 20 mV < ζ < −20 mV at pH < 7–8. Full coverage of the surface of nanoparticles with humic acids was assessed by the zeta potential value, ζ > −20 mV, over the entire pH range.

The nanoparticles IEP shifted towards low pH with an increasing HP concentration (from ~6.2 at 0 g/g HP to ~3 at the maximum concentration of HP). For all samples, a shift in the isoelectric point was also observed, indicating a straight correlation of the degree of nanoparticle coverage with the HP (Figure 5). At the position of the IEP corresponding to a pH of ~3 or less, the MNP surface was entirely concealed with the humic preparation, whereas at about pH ~6–7, the surface of these nanoparticles was practically uncoated upon humic preparation.

According to [48], the presence of functional groups of amino-silica provides adsorption sites on the NP surface for organic or inorganic compounds. Deposition of the silica on the surface of Fe_3_O_4_ NPs leads to an increase in the S_BET_ value, for example, from 114 m^2^·g^−1^ to 216 m^2^·g^−1^, respectively, for bare and modified MNPs [53]. Moreover [48], the increase in surface area can be explained by an increase in the number of functional groups (both -OH and amino groups), as well as by an increase in the total pore volume upon APTES modification. A variety of combinations of functional groups was shown in our earlier publication [40], as reported in Figure 2. In addition, the increase in surface area is in good agreement with the enhancement in sorption capacity of functionalized magnetite compared to bare magnetite in relation to humic acids [53], respectively from 0.039 g/g to 0.052 g/g. The low Fe_3_O_4_/APTES sorption capacity was probably associated with the presence of a large number of impurities and the screening of NH_3_^+^ onto the surface, and, thus, in order to remove the impurities a further process, i.e., dialysation, was performed.

Sample preparation conditions have an impact on the sorption capacity, in fact by using dialysis, the Fe_3_O_4_/APTES sample showed an improved capability for adsorbing larger amounts of HP compared to the non-dialyzed samples. This result could be analyzed as likely due to the amount of APTES functional groups available for binding, upon humic preparation, increasing due to the impurity removal when using the dialysis process. Moreover, the MNPs, exposed to the magnetic stirring (Fe_3_O_4_/APTES (S)), have a larger sorption capacity than the sample upon ultrasonic bath (Fe_3_O_4_/APTES (Us)), likely due to the harsh conditions associated with sonication (i.e., extremely high temperature and pressure in cavities forming at interfaces), which are capable of destroying the Fe_3_O_4_/APTES-HP bonds and thus altering the original loading capacity.

#### 3.2.4. Colloidal Stability of HP-Coated MNPs

A change in the surface state of nanoparticles due to HP adsorption on Fe_3_O_4_/APTES NPs was demonstrated by changes in zeta potential (Figure 6), as a function of the added amount of HP (pH~5 and KCl = 0.01 M).

As can be appreciated in Figure 6, the addition of a specific amount of HP led to a decrease in zeta potential to zero (i.e., the MNP surface charge was fully neutralized). The HP amounts required to reach this specific state of NPs were definitely different, especially for the Fe_3_O_4_/APTES (S) sample.

Figure 7 shows a change in the colloidal stability of MNPs at pH~5 and KCl = 0.01 M. In an HP-free system, the low charge density and low electrokinetic potential of the Fe_3_O_4_/APTES NPs led to their aggregation and sedimentation. At the same time, the Fe_3_O_4_/APTES (S) and the Fe_3_O_4_/APTES (Us) samples remained stable at pH 5 without HP.

A certain concentration of adsorbed HP led to an NP charge of approximately −20 mV and stabilized the system, to form of a transparent colloidal sol. Samples of Fe_3_O_4_/APTES and Fe_3_O_4_/APTES (Us) with the humic preparation concentrations of 0–0.02 became unstable, aggregated, and sedimented. However, the Fe_3_O_4_/APTES (S) sample did not demonstrate aggregation at a low concentration of HP, up to 0.02 g/g, and became stable when 0.038 g/g of HP was added.

The experimental values of the amount of HP to change the surface charging of the initial and treated MNPs are summarized in Table 3.

The adsorption of HP led to the inversion of the zeta potential, from a positive to negative sign. Increasing to −20 mV with a different amount of adsorbed HP (Table 3). While, HP adsorption increased up to a plateau value (different for various NPs), the (negative) zeta potential of the particles changed negligibly with increasing amounts of added HP (Table 3), likely due to the adsorption saturation. Moreover, this was also expected, due to the limited increase in zeta potential and due to the condensation of counter ions at high potentials and low electrolyte concentrations [54]. The loading capacity of HP (max value for Fe_3_O_4_/APTES (S)) was considerably higher than the electrostatic charge compensation point, probably due to the increase in the HP density of the adsorbed layer, which was accompanied by counter ion adsorption to overcome the electrostatic repulsion among COO−groups within the loops of the adsorbed polyelectrolyte.

When humic preparations were added, the hydrodynamic size depended on the amount and position of the isoelectric point, which moved to the left. For all nanoparticles, the average hydrodynamic size decreased with an increasing volume of humic preparation and the level of MNP surface coverage (Figure 3). In the case of partially coated NPs, the average hydrodynamic size depended on pH, whereas for fully covered nanoparticles, the average hydrodynamic size was independent of pH and held constant. The value of the maximum size correlates, absolutely, with the position of the isoelectric point at which the system is unstable. The average hydrodynamic size of the initial sample of Fe_3_O_4_/APTES was 300 nm near the IEP. This value decreased with an increasing amount of humic preparation. When the MNP surface was fully covered with HP, the average hydrodynamic size became ca. 150 nm. The addition of 0.026 g of humic preparation led to a change in the average hydrodynamic diameter, close to the isoelectric point, varying from 329 nm for Fe_3_O_4_/APTES to 654 nm for Fe_3_O_4_/APTES-S. For pH~4 and the same quantity of HP, the value of the hydrodynamic diameter for all samples was within a range of 150–300 nm. After an increase in the amount of HP added, the average hydrodynamic diameter remained almost unchanged for all samples. By adding the maximum volume of humic preparation for Fe_3_O_4_/APTES and Fe_3_O_4_/APTES-S, the average hydrodynamic diameter became 100–200 nm for all nanoparticles.

## 4. Conclusions

In this study, using silica modified nanoparticles Fe_3_O_4_/APTES as a model system, we explored two techniques for the dispersion of pre-formed nanoparticle suspensions into stable fractions, according to zeta potentials and hydrodynamic diameters. Two different techniques for dispersing silica MNPs in terms of their zeta potential and hydrodynamic diameter were reported and compared. Magnetic stirring was revealed to provide a good approach for the oxygen-sensitive dispersion of iron-containing nanoparticles, and it was efficient in terms of the loading capacity of functional humic substances, due to surface charging. The sonication-based method resulted in a low loading capacity, likely due to high temperature and pressure in the cavities formed at the interfaces, which induced breakage of the silica shell. The achieved data also confirmed that a electrophoretic mobility technique can be applied to predict the loading capacity for polyelectrolytes such as humic preparations.

## Figures and Tables

**Figure 1 nanomaterials-11-03295-f001:**
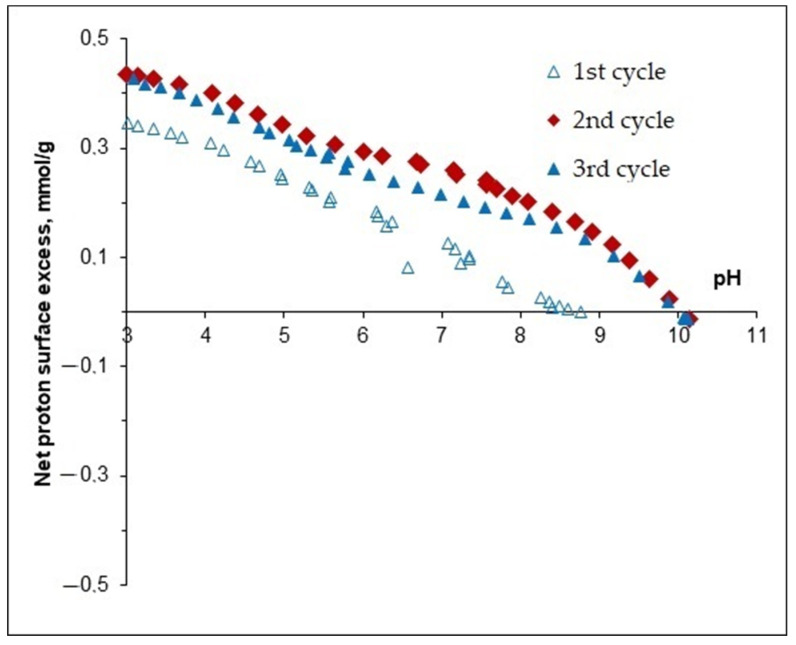
pH-dependent surface charging of Fe_3_O_4_/APTES at different 0.005 M KCl concentrations. The experimental points were calculated from the material balance of H+/OH− during equilibrium acid-base titration.

**Figure 2 nanomaterials-11-03295-f002:**
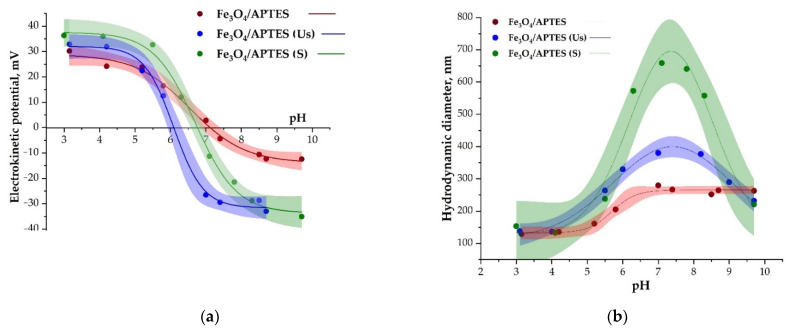
Zeta potential (**a**-left) and hydrodynamic diameter (**b**-right) of the Fe_3_O_4_/APTES samples after dialysis–sonication and dialysis–stirring as a function of pH (0.01 M KCl).

**Figure 3 nanomaterials-11-03295-f003:**
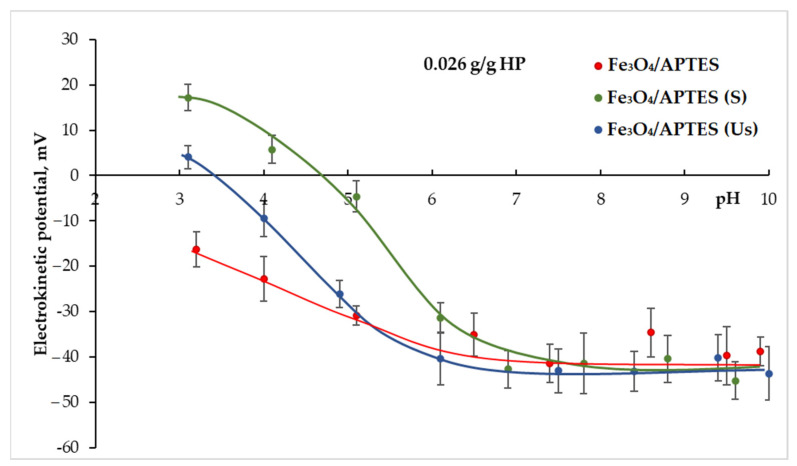
Effect of humic preparation (0.026 g/g) adsorption on the pH-dependent zeta potential of samples ± standard deviation.

**Figure 4 nanomaterials-11-03295-f004:**
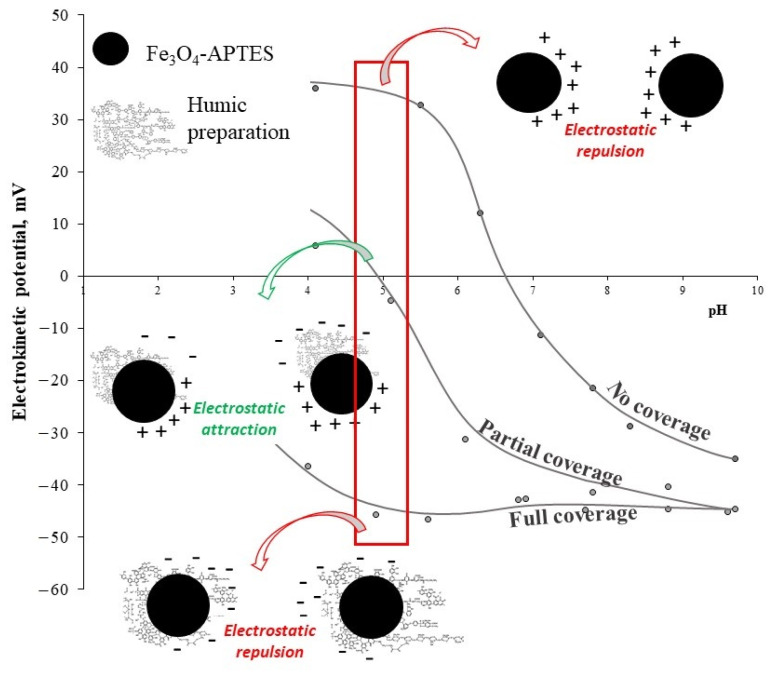
Schematic representation of mechanism of HP effect on the MNP colloidal stability.

**Figure 5 nanomaterials-11-03295-f005:**
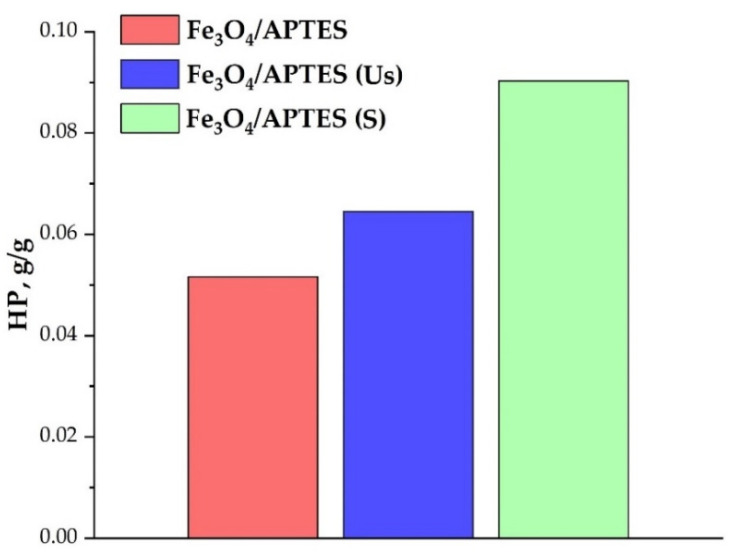
Amount of HP for a complete coverage of 1 g MNPs surface given in g (ζ~−20 mV, pH 3, 0.01 M KCl).

**Figure 6 nanomaterials-11-03295-f006:**
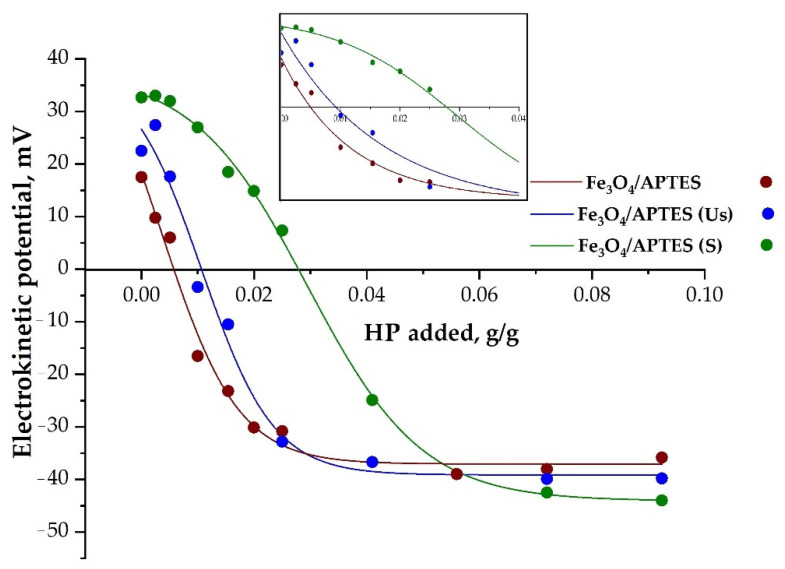
Effect of HP on the surface charging of Fe_3_O_4_/APTES samples (pH ~5 and KCl = 0.01 M). The curves were drawn as guidelines by eye.

**Figure 7 nanomaterials-11-03295-f007:**
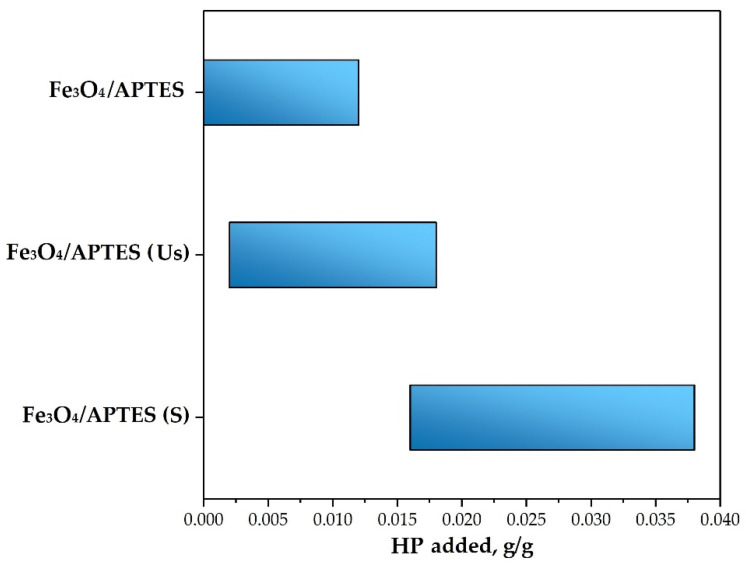
System instability interval depending on the HP quantity (pH~5 and KCl = 0.01 M). In the intervals corresponding to the blue markers, the samples coated with humic preparation at these concentrations had a zeta potential value of −20 to 20 mV and were unstable (they tended to aggregate and sediment.).

**Table 1 nanomaterials-11-03295-t001:** Microstructure of MNPs.

Sample	Fe_3_O_4_	Fe_3_O_4_/APTES
*a*, A	8.3813	8.3789
Structure	Fe_2.93_O_4_	Fe_2.88_O_4_
% Fe_3_O_4_	78.8	63.7
D_XRD_, nm	17.1 ± 2.3	20.5 ± 3.3
CV, %	13.5	16.1

*a*-unit cell parameter, Å; %-content of magnetite in magnetite/maghemite compound; D_XRD_ is average particle size calculated by the Scherrer equation ± standard deviation, nm; CV is coefficient of variation characterizing the polydispersity of the system, %.

**Table 2 nanomaterials-11-03295-t002:** Electro-kinetic parameters of MNPs.

Sample	Fe_3_O_4_/APTES	Fe_3_O_4_/APTES (Us)	Fe_3_O_4_/APTES (S)
IEP (ζ = 0)	7.1	6.3	6.6
Max ζ-potential, mV	30.2 ± 7.2	32.9 ± 6.1	36.3 ± 7.5
Min ζ-potential, mV	−12.4 ± 5.5	−33.1 ± 5.2	−35.1 ± 5.1

**Table 3 nanomaterials-11-03295-t003:** Amount of HP needed to change the surface charging of the Fe_3_O_4_/APTES samples.

HP Amount, g/g	Fe_3_O_4_/APTES	Fe_3_O_4_/APTES (Us)	Fe_3_O_4_/APTES (S)
for full neutralization of charge	0.004	0.01	0.025
to achieve −20 mV of zeta potential	0.014	0.018	0.038
to reach plateau	0.04	0.028	0.072

## Data Availability

The data presented in this study are available on request from the corresponding author.

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
