# Peer review of "Colloidal Stability of Silica-Modified Magnetite Nanoparticles: Comparison of Various Dispersion Techniques"

_nanomaterials, 2021, doi:10.3390/nano11123295_

Round 1

Reviewer 1 Report

The authors focused on the effect of two different dispersion means, ultrasonic means and magnetic stirring, on the colloidal stability of ferric tetroxide nanoparticles after silica encapsulation, using their nanoparticles. The samples were characterized by XRD, zeta potential, and dynamic light scattering techniques, and the results showed that the magnetic stirring method was superior to the ultrasonic dispersion.

From my personal point of view, the authors obtained more useful information on the enhancement of the colloidal dispersion of silica-coated ferric tetroxide nanoparticles and gave a more reasonable explanation for this enhancement of colloidal stability. Moreover, the article is better written and organized. Therefore, I recommend this article particles to be considered for publication in this journal after minor revision. Some comments are as follows.

  1. i) The dispersion of the particles and the intrinsic size of the particles are also very relevant, can the authors provide the information on the size morphology of the iron tetroxide particles.

2) Ultrasonic mode now has greater energy intensity to provide, which is helpful to break up particle agglomerates. However, too strong energy injection may damage the surface or internal structure of the particles themselves. Can the authors provide a discussion on this piece, such as whether it is possible to balance these two for optimization purposes.

Author Response

Dear Reviewer,

Thank you for your valuable and detailed comments regarding this article.

Below you will find answers to your comments.

Reviewer 2 Report

This paper compares the effect of ultra sonication and simple magnetic stirring on the stability of Fe2O3 particles coated with a layer a silica with or without an absorbed layer of poly anions. The paper presents several characterizations, however I have a number of concerns, as I explain below. 

1) The authors refer to the deposition of a silica layer onto Fe2O3 using without distinction the terms  "grafting with silane", "functionalization with SiO2", "coating with SiO2-NH2", and "Stöber method". These are different things. Grafting with silane implies monolayer coverage by covalent bonds of a silane sot hat the particle surface contains blushes of the tail of the silane. "Functionalization" also implies grafting, so the term "functionaization with SiO2" is not correct. "Coating with "SiO2-NH2" is an unknown process to me. The "Stöber method" is a recipe for synthesizing monodisperse silica from alkoxide, not silanes. This misrepresentations must be cleaned up. 

2) The XRD analysis of Fe2O3 is superfluous and should be removed. The authors are using a recipe from the literature, therefore the properties of the Fe2O3 particles should be are those expected for that recipe. The authors should validate that their implementation of the recipe gives the expected results but this validation is secondary to the goals of the work and should not be given a page of the manuscript. 

3) Table 1 is unclear and confusing. None of the acronyms on the table have been explained and their relevance has not been discussed.

4) On Table 1 the authors give the SEM size of the particle but in the text, the SEM size is cited from a paper from the same group (Ref 33).  If these particles have been characterized elsewhere, what is the point of this work?

5) On page 5, line 176 the authors state: 

"The particle diameter upon APTES modification increased up to 20.5 nm and 24.18 nm as compared to bare magnetic nanoparticles."

They apparently refer to to the values D(XRD) and D(SEM) in Table 1. What size is this? The hydrodynamic diameters in Figs 4 and 5 show particle size of the order of 150nm and above. Are the authors implying that these are composed of 20-25 nm primary particles? If that's the case I cannot see how XRD can give that size. 

Moreover, why would XRD give a different size for the silica coated particles? Silica is amorphous and does not show up in XRD. 

6) Figure 5 is too busy and also confusing due to the facts that the vertical axes are not the same fir US and S samples, and that the S sample is missing the ratio 0.007, which makes one-to-one comparisons difficult. Regardless, the size measurements seem redundant because they simply state that the dispersion is stable once its zeta is sufficiently negative. 

7) The main point of the paper is to examine the effect of ultra sonication over magnetic stirring. Ultra sonication seems to produce poorer results. The authors attribute those to "extremely high temperature and pressure in cavities forming at interfaces,  capable of destroying the Fe3O4/APTES-HP bonds and thus altering the 319 original loading capacity". This is speculation. Is there some evidence for this damage? 

Unfortunately the authors never explain the purpose of sonication/stirring. Apparently this is done to disperse the particles after they have been coated with silica, but is it because the coating process itself causes aggregation, is it because the uncoated Fe2O3 is aggregated already, or is it both?

In summary, I find the paper leaves several important questions unanswered and leaves an unclear sense as to what conclusions one may draw from these results.

Author Response

(The authors gave the same response as above.)

Reviewer 3 Report

The manuscript contains an analysis of a kind of nanoparticles of certain potential interest, as magnetite/silica. It needs, however, in my opinion, significant improvement in order to be acceptable for Namomaterials. These are, in order of appearance in the text, the points that might be considered by the authors:

  1. The title is incorrectly spelled: ..of the silica modified... -> ...of silica modified
  2. In the Introduction it is mentioned that magnetic NPs are of interest in drug delivery or other biomedical aspects; however, nothing is mentioned about the necessity fo silica coating
  3. It is also mentioned that MR fluids are useful in airplane landing gear damping. There are many otehr applications, and I doubt that they are really used in such a demanding system as the landing gear
  4. The work is focussed on humic substances adsorption, but nothing is said in the Introduction regarding the reasons for this selection. 
  5. Page 3: frequency should be kHz, not Hz; also, he power used must be specified
  6. By the way, how can you sue a magnetic stirrer with magnetic particles that end up adhered to the magnetic bar?
  7. Line 146 page 4: how significant is the difference between lattice parameters (8.38 vs. 8.37; what is the uncertainty ±?)
  8. Table 1. Significant figures: 32.1+-4.33 -Z 32.1 +-4.3, etc.
  9. The titrations in Figs. 2,3 are well known methods and results. These Figures do not provide new results, and they shpuld be deleted from the manuscript
  10. Line 216 page 6: use of sonication in EU projects is not  a scientific argument. and additionally the technique is a very classical one
  11. Figure 4 (and the rest of the paper): I do not see significant difference sbetwen sonicating and stirring. All this discussion is not of interest presently
  12. The symbol for the zeta potential is wrong (ξ instead of ζ) in many places
  13. Very important: the zeta potential is meaningless in polyelectrolyte-coated particles because a slip plane cannot be identified (c.f. works by H. Ohshima). Only the mobility can be used
  14. Table 2: please indicate uncertainties (±)
  15. Fig. 5 confirms essentially that there is no need to include any discussion on the difference between sonicating or stirring. I would include just one set of data
  16. Line 303 page 11: I cannot see why the number of functional groups can affect the surface area. Please explain
  17. I cannot understand Fig. 9. Probably a much more clear caption is needed

Author Response

(The authors gave the same response as above.)

Round 2

Reviewer 3 Report

The authors have considered all the observations that I made on my previous revision. I think that the paper can now be accepted.